# Nano DNA Vaccine Encoding *Toxoplasma gondii* Histone Deacetylase SIR2 Enhanced Protective Immunity in Mice

**DOI:** 10.3390/pharmaceutics13101582

**Published:** 2021-09-29

**Authors:** Zhengqing Yu, Yujia Lu, Wandi Cao, Muhammad Tahir Aleem, Junlong Liu, Jianxun Luo, Ruofeng Yan, Lixin Xu, Xiaokai Song, Xiangrui Li

**Affiliations:** 1Ministry of Education (MOE) Joint International Research Laboratory of Animal Health and Food Safety, College of Veterinary Medicine, Nanjing Agricultural University, Nanjing 210000, China; 2018207044@njau.edu.cn (Z.Y.); 11118310@njau.edu.cn (Y.L.); 17119427@njau.edu.cn (W.C.); 2018207076@njau.edu.cn (M.T.A.); yanruofeng@njau.edu.cn (R.Y.); xulixin@njau.edu.cn (L.X.); songxiaokai@njau.edu.cn (X.S.); 2State Key Laboratory of Veterinary Etiological Biology, Key Laboratory of Veterinary Parasitology of Gansu Province, Lanzhou Veterinary Research Institute, Chinese Academy of Agricultural Sciences, Lanzhou 730046, China; liujunlong@caas.cn (J.L.); luojianxun@caas.cn (J.L.)

**Keywords:** *Toxoplasma gondii*, histone deacetylase, SIR2, chitosan, PLGA, immune protection

## Abstract

The pathogen of toxoplasmosis, *Toxoplasma gondii* (*T. gondii*), is a zoonotic protozoon that can affect the health of warm-blooded animals including humans. Up to now, an effective vaccine with completely protection is still inaccessible. In this study, the DNA vaccine encoding *T. gondii* histone deacetylase SIR2 (pVAX1-SIR2) was constructed. To enhance the efficacy, chitosan and poly (d, l-lactic-*co*-glycolic)-acid (PLGA) were employed to design nanospheres loaded with the DNA vaccine, denoted as pVAX1-SIR2/CS and pVAX1-SIR2/PLGA nanospheres. The pVAX1-SIR2 plasmids were transfected into HEK 293-T cells, and the expression was evaluated by a laser scanning confocal microscopy. Then, the immune protections of pVAX1-SIR2 plasmid, pVAX1-SIR2/CS nanospheres, and pVAX1-SIR2/PLGA nanospheres were evaluated in a laboratory animal model. The in vivo findings indicated that pVAX1-SIR2/CS and pVAX1-SIR2/PLGA nanospheres could generate a mixed Th1/Th2 immune response, as indicated by the regulated production of antibodies and cytokines, the enhanced maturation and major histocompatibility complex (MHC) expression of dendritic cells (DCs), the induced splenocyte proliferation, and the increased percentages of CD4^+^ and CD8^+^ T lymphocytes. Furthermore, this enhanced immunity could obviously reduce the parasite burden in immunized animals through a lethal dose of *T. gondii* RH strain challenge. All these results propose that pVAX1-SIR2 plasmids entrapped in chitosan or PLGA nanospheres could be the promising vaccines against acute *T. gondii* infections and deserve further investigations.

## 1. Introduction

*Toxoplasma gondii* (*T. gondii*) is an obligate intracellular parasite distributed worldwide and infect a vast array of warm-blooded vertebrates including humans [1,2]. *T. gondii* infections occurred in healthy individuals are asymptomatic, but severe consequences such as abortion and encephalitis can be posed to pregnant women and immunocompromised patients [3,4]. Toxoplasmosis can be transferred through contaminated water, meats, fruits, and even vegetables [5,6,7]; this feature made *T. gondii* one of the most important zoonotic pathogens and emphasizes the necessity of intervention [8]. To date, effective vaccines against *T. gondii* still remain unavailable. Although therapeutic drugs are possible [9], their application are limited by its toxicity and adverse reactions [10]. Thus, prophylactic strategies are considered to be preferred in *T. gondii* preventions [11]. Based on the *T. gondii* S48 strain, a live-attenuated vaccine, Ovilis Toxovax^®^ (Intervet Inc., Angers, France), has been licensed for the use in parts of countries that prevent abortion in sheep from congenital toxoplasmosis [12]. However, it is inadequate and the protective immunity in tissue cyst formation is still under question [13,14]. Due to the major transmission routes in *T. gondii*, it is a valued and urgent requirement to develop an effective vaccine against *T. gondii* infections.

Owing to the high-risk factors of *T. gondii* infections on a global scale, development of vaccine pipelines and immunotherapies is a great priority in toxoplasmosis prevention [15]. For this purpose, it is of urgent necessity to find promising vaccine targets with high immunogenicity [16,17]. An effective target should have high immunogenicity, low-allergic, and relatively non-toxic characteristics. In addition to dense granule (GRAs) proteins [18], surface antigens (SAGs) [19], rhoptries (ROPs) [20], and micronemes (MICs) [21], *T. gondii* have evolved plenty of additional proteins for a successful duplication in host cells [22]. To date, a wide range of antigens have been used in development of vaccine against toxoplasmosis, but an effective vaccine is still unavailable [23,24]. As is known, *T. gondii* has two distinct asexual life cycle in mammalian hosts, bradyzoite and tachyzoite, and each developmental stage of the parasite is distinct in morphology and biochemistry, leading to the adaptation to different environments in different hosts [25]. Development transition of *T. gondii* is often associated with gene expression changes [26], and gene expression of *T. gondii* is promoted by epigenetic events. Interests have been greatly aroused with the discovery of compounds that can inhibit the activities of histone-modifying enzymes [27,28]. Acetylation is the main post-translational modification in histone proteins [29,30,31], and histone deacetylases (HDACs) and histone acetylases (HATs) play an important role in epigenetic regulation as well as intracellular process [32]. Furthermore, by changing the structure of histone proteins, these post-translational modifications are also involved in chromatin formulation.

Related to gene transcription, gene expression, and chromatin remodeling, histone remodelers are important in cell replications, prompting an increasing interest in HDACs and HATs used as therapeutic targets [33,34]. Inhibition of HDAC activity has been proved effective in suppressing cell proliferation [35], and a similar effect was also validated in *T. gondii* tachyzoite [36]. Based on the published paper, four HDAC homologues (Class I/II) as well as one SIR2 subtype homologue (class III) have been detected in *T. gondii* genome [37]. Due to the regulating role in virulence gene expression [38,39], the SIR2 gene plays an essential role in some protozoan. According to the published paper, recombinant *T. gondii* SIR2 protein exhibits a good capacity in promoting proliferation, phagocytosis, and nitric oxide (NO) secretion in vitro [40]. Furthermore, the SIR2 protein has demonstrated an important role in virulence gene regulation, telomere length maintenance, and heterochromatin formulation [38,39,41]. Thus, development of *T. gondii* vaccine targeting the SIR2 protein would be a promising way to prevent toxoplasmosis.

Widely used in different fields of science, nanotechnology has aroused increasing attention in triggering adequate immunity against *T. gondii* [42]. Nowadays, many synthetized nanospheres have shown its superior performance in sustained release and anti-degradation [43]. Characterized by biodegradability, biocompatibility, and relative safe nature [44], chitosan is a natural polysaccharide deacetylated from chitin in alkaline conditions [45]. In addition, as a promising delivery system for vaccines and drugs, chitosan nanospheres are allowed its applications in wound dressings by FDA [46] and have been proved to be safe in dietary applications [47]. Approval by the Food and Drug Administration (FDA), poly (d, l-lactic-*co*-glycolic)-acid (PLGA) is widely used in vaccine and drug delivery due to its non-toxicity, tissue biocompatibility, and biodegradability [48]. As an attractive carrier and adjuvant, chitosan and PLGA have been evaluated to be suitable in delivery of recombinant protein or DNA vaccine [49,50]. Herein, we made a new attempt to synthesize chitosan and PLGA nanospheres based on the pVAX1 plasmids expressing *T. gondii* histone deacetylase SIR2 (pVAX1-SIR2). The physical characteristics including release characteristics were analyzed, and the immune protections of chitosan and PLGA nanospheres were then investigated in animals to develop a DNA vaccine against acute *T. gondii* infections.

## 2. Materials and Methods

### 2.1. Cells, Parasites, and Animals

Human embryonic kidney 293-T (HEK 293-T) cells were preserved in the Ministry of Education (MOE) Joint International Research Laboratory of Animal Health and Food Safety, College of Veterinary Medicine, Nanjing Agricultural University, Nanjing, PR China, and cultured in Dulbecco’s Modified Eagle’s Medium (DMEM, Gibco, Carlsbad, CA, USA) containing 100 U/mL penicillin (Carlsbad, Gibco, CA, USA), 100 mg/mL streptomycin (Carlsbad, Gibco, CA, USA), and 10% fetal bovine serum (FBS, Gibco, Carlsbad, CA, USA). HEK 293-T cells were kept at 37 °C in a 5% CO_2_ atmosphere.

The tachyzoites of *T. gondii* type I strain (RH) were also provided by the Ministry of Education (MOE) Joint International Research Laboratory of Animal Health and Food Safety, College of Veterinary Medicine, Nanjing Agricultural University, Nanjing, PR China. To maintain and reproduce the parasites, the previous method was conducted [51] in specific pathogen free (SPF) BALB/c mice.

SPF BALB/c mice weighting 18–22 g were purchased from the Center of Comparative Medicine, Yangzhou university, China, and were strictly housed in SPF environment. The Sprague Dawley (SD) rats weighting 200–220 g were obtained from the Model Animal Research Center (MARC), Nanjing University, China, and kept in the same environment with BALB/c mice.

### 2.2. Construction of the Recombinant Plasmid

Based on the instructions of total RNA extraction kit (OMEGA Bio-tek, Norcross, GA, USA), 10^6^ tachyzoites of *T. gondii* RH strain was used for RNA isolation. To synthesize the cDNA, the reverse transcription PCR (RT-PCR) was conducted by using the reverse transcription kit (Takara Biotechnology, Dalian, China). The well-prepared cDNA was stored at −80 °C until use. Targeting the conserved domain sequences (CDS) of *T. gondii* histone deacetylase SIR2 (GenBank: XM_002366314.2), the forward and reverse primers, 5′-CCG **GAATTC** GCCACC ATGAACCTTTTGGTTATTTCG-3′ and 5′-CCG **CTCGAG** CTATCTCCTTTGCAGCGC-3′, were designed along with the Kozak translation initiation sequence (underlined) and the restriction endonuclease sites (*Eco*RI and *Xho*I, marked in bold). The designed primers were synthesized by Tsingke biological technology (Nanjing, China). To obtain the PCR amplicons, 12.5 µL 2 × Phanta Max Master Mix (Vazyme, Nanjing, China), 10 pmol of each primer, 1 ng cDNA template, and ddH_2_O were involved to make a final volume of 25 µL in each reaction tube, and the following program was conducted in an amplifier (Thermo Scientific, Waltham, MA, USA): preheating (5 min at 95 °C), amplification for 35 cycles of 30 s at 95 °C, 30 s at 54.2 °C, and 70 s at 72 °C, and the final extension (5 min at 72 °C). The amplicons were purified by gel extraction kit (OMEGA Bio-tek, Norcross, GA, USA), after visualization in 2.0% agarose gel containing 0.01% nucleic acid dye (Ultra GelRed, Vazyme, Nanjing, China). Digested with restriction endonuclease (Takara Biotechnology, Dalian, China), the obtained amplicons were ligated to the enzyme-digested pVAX1 vector (Invitrogen Biotechnology, Shanghai, China) by T4 DNA ligase (Takara Biotechnology, Dalian, China). The constructed vector was identified by double enzyme digestion analysis and was confirmed by ABI PRISM™ 3730 XL DNA Analyzer (Applied Biosystems, Waltham, MA, USA). The confirmed pVAX1-SIR2 plasmids were transferred to *Escherichia coli* (*E. coli*) DH5α strain (Vazyme, Nanjing, China).

To obtain a prokaryotic expression plasmid, the same methods were conducted as mentioned above. The restriction endonuclease site in reverse primer was changed to *Hind*III (Takara Biotechnology, Dalian, China); thus, the reverse primer was redesigned (5′-CCC AAGCTT CTATCTCCTTTGCAGCGC-3′). The purified amplicons were ligated to the linearized pET-32a vector (Invitrogen Biotechnology, Shanghai, China) using the same methods as described above. Analyzed by DNA sequencing, the confirmed recombinant pET-32a plasmid was transferred to *E. coli* BL21 (DE3) strain (Vazyme, Nanjing, China).

### 2.3. Plasmid Preparation and Transfection In Vitro

To prepare the pVAX1-SIR2 plasmids in large quantities, *E. coli* DH5α containing the pVAX1-SIR2 plasmid was reproduced in Luria Bertani (LB) medium containing 100 μg/mL kanamycin for 16 h at 37 °C and shaking at 180 rpm. The pVAX1-SIR2 plasmids were purified by the endofree plasmid kit (Vazyme, Nanjing, China), and the endotoxin was detected by ToxinSensor™ Chromogenic LAL Endotoxin Assay Kit (GeneScript, Piscataway, NJ, USA). For concentration determination, the nanodrop microvolume spectrophotometer (NanoDrop 3000, Thermo Scientific, Waltham, MA, USA) was used. The pVAX1-SIR2 plasmids were stored at −20 °C until use.

The pVAX1-SIR2 plasmid expression was determined by transfection of HEK 293-T cells with the Lipofectmine™ 3000 reagent (Invitrogen Biotechnology, Shanghai, China) according to the instructions. Briefly, HEK 293-T cells were cultured in a six-well plate (Costar, Cambridge, MA, USA) with 80% confluence, then the plasmid-lipid complex was added based on the guidelines of instructions. Then, the cells were cultured for three days without changing cell culture medium.

### 2.4. Generation of Anti-rTgSIR2 Polyclonal Antibodies

To obtain the polyclonal antibodies against recombinant *T. gondii* SIR2 protein (rTgSIR2), the *E. coli* BL21 (DE3) was induced to express rTgSIR2 by isopropyl β-d-thiogalactoside (IPTG, Sangon Biotech, Shanghai, China). Purified by nickel-affinity chromatography column (GE Healthcare, Marlborough, MA, USA), the eluted rTgSIR2 was analyzed by 12% SDS-PAGE electrophoresis and quantified by BCA protein assay kit (Bio-Rad, Hercules, CA, USA). The purified rTgSIR2 was stored at −80 °C until use. To generate polyclonal antibodies against rTgSIR2, SD rats were subcutaneously vaccinated with 200 μg of rTgSIR2 emulsified with an equal volume of complete Freund’s adjuvant (Sigma, Saint Louis, MO, USA). Fourteen days later, booster vaccinations were conducted with 200 μg of rTgSIR2 emulsified with an equal volume of incomplete Freund’s adjuvant (Sigma, Saint Louis, MO, USA) through the subcutaneous injection. The booster vaccinations were performed five times in total and the time interval between two vaccinations was seven days. Seven days after the last vaccination, the blood was collected at the eye socket and the serum was harvested.

### 2.5. Immunofluorescence Staining

As described in Section 2.3, the transfected HEK 293-T cells were obtained after a three-day culture. Cells were washed with phosphate-buffered saline (PBS) for three times and fixed with 1 mL/well 4% paraformaldehyde for 12 h at 4 °C. On a rotary shaker at 50 rpm, cells were immerged in tris buffered saline (TBS) containing 0.1% Triton X-100 (TBSx) for 3 min. Then, cells were subsequently incubated at 37 °C with TBSx containing 5% BSA and 1% serum from the rTgSIR2-vaccinated rat for 1.5 h on a rotary shaker at 50 rpm. After washing three times in TBSx, cells were then incubated at 37 °C in TBSx containing 5% BSA and 0.2% CY3-conjugated anti-rat IgG (Sigma, Saint Louis, MO, USA) on a rotary shaker at 50 rpm. After washing three times in TBSx again, 500 μL of 4′,6-diamidino-2-phenylindole (DAPI) staining solution (Beyotime, Shanghai, China) were added into each well and incubated at 37 °C on a rotary shaker at 50 rpm. Coverlips were mounted with 300 μL anti-fade mounting medium (Beyotime, Shanghai, China) after DAPI staining solution was completely removed. Cells were then immediately imaged by a laser scanning confocal microscopy (Nikon A1 plus, Nikon Corporation, Tokyo Metropolis, Japan).

### 2.6. Nanospheres Formulation

With minor modifications, double emulsion evaporation technique (*w/o/w*) was conducted in synthesis of PLGA nanospheres according to the previous study [52]. Briefly, 5 mg of PLGA (MW: 40,000–75,000 Da, LA/GA: 65/35, Sigma, Saint Louis, MO, USA) was firstly dissolved in 1 mL dichloromethane (DCM, Sigma, Saint Louis, MO, USA) to form the organic phase. To construct the *w/o* emulsion, the organic phase was sat on a magnetic stirrer (400 rpm) at room temperature, and the 2 mL 5% (*w/v*) polyvinyl alcohol (PVA, MW: 31,000–75,000 Da, Sigma, Saint Louis, MO, USA) was dropped in. Subsequently, in an ice bath, tip sonication (Scientz Biotechnology, Ningbo, China) was conducted in a continuous mode for 3 s at 3 s (10 min in total) under the output power of 30 W, until the color of aqueous phase turned milky white. Then, 4 mL purified plasmids (1 mg/mL) were added dropwise and sonicated using the same method as described above. To construct the *w/o/w* emulsion, 2 mL 5% (*w/v*) PVA was dropwise dissolved in *w/o* emulsions, and tip sonication was again carried out. To harvest PLGA nanospheres, the *w/o/w* emulsions were centrifuged at 40,000 rpm for 20 min at 4 °C, then the precipitation were collected, redissolved in double-distilled water, passed through a 0.22 μm filtering membrane (Millipore, Billerica, MA, USA), and frozen at −80 °C for at least 2 h. To completely remove DCM, the frozen nanospheres were quickly transferred into a vacuum freeze dryer (Labconco, Kansas, MO, USA) until thoroughly freeze-dried. The PLGA nanospheres with plasmids and non-plasmid were prepared together. The well-prepared PLGA nanospheres were stored in airtight vials at −20 °C until use.

As for chitosan nanospheres, the ionic gelation technique was carried out for preparation of chitosan nanospheres following the previous study with minor modifications [53]. In brief, 1% (*v/v*) aqueous solution of acetic acid was first prepared, and 100 mg chitosan (MW 50–190 kDa, Sigma, Saint Louis, MO, USA) were dissolved in 50 mL acetic acid solution to form 0.2% (*w/v*) chitosan solution. To make chitosan fully dissolved, the chitosan solution was sat on a rotary shaker (180 rpm) at room temperature for at least 30 min. Then, the pH value was adjusted to 5.0 by 2 M NaOH solution. To obtain 2 mg/mL sodium tripolyphosphate (TPP, Aladdin, Shanghai, China), 20 mg TPP was dissolved in 10 mL double-distilled water. Under room temperature, 4 mL TPP and 4 mg (the concentration was 1 mg/mL) purified plasmids were subsequently dropped into 20 mL chitosan solution, which was sat on a magnetic stirrer (400 rpm). Under the output power of 50 W in an ice bath, tip sonication was conducted in a continuous mode for 3 s at 3 s (2 min in total). Centrifuged at 40,000 rpm for 20 min at 4 °C, the chitosan nanospheres were collected, redissolved in double-distilled water, and passed through a 0.22 μm filtering membrane. Before transferred into the vacuum freeze dryer, the chitosan nanospheres were freezed at −80 °C for at least 2 h. The nanospheres were then thoroughly freeze-dried and stored in airtight vials at −20 °C until use. The chitosan nanospheres with plasmids and non-plasmid were both synthesized.

### 2.7. Characterizations and Release Characteristics of Nanospheres

To analyze the features of PLGA and chitosan nanospheres, the freeze-dried nanospheres were sent to College of Life Science, Nanjing Agriculture University for scanning electron microscope (SEM) analysis (SU8010, Hitachi, Tokyo, Japan). By arbitrarily measuring five nanospheres, the average diameter of PLGA and chitosan nanospheres was evaluated by ImageJ software version 1.8 (NIH Image, Bethesda, MD, USA).

To calculate the encapsulation efficiency (EE) and the loading capacity (LC), the concentration of unbound plasmids in the supernatant after ultracentrifugation (described in Section 2.6) was quantified by the nanodrop microvolume spectrophotometer referenced to non-plasmid supernatant. The EE and LC were then evaluated by Formulas (1) and (2), respectively. Determination of EE and LC were conducted three times.
(1)$$\mathrm{EE}\;\left(\%\right)=\frac{\mathrm{Total}\;\mathrm{plasmid}-\mathrm{Free}\;\mathrm{plasmid}}{\mathrm{Total}\;\mathrm{plasmid}}\times 100\%$$
(2)$$\mathrm{LC}\;\left(\%\right)=\frac{\mathrm{Total}\;\mathrm{plasmid}-\mathrm{Free}\;\mathrm{plasmid}}{\mathrm{Weight}\;\mathrm{of}\;\mathrm{nanospheres}}\times 100\%$$

The release characteristics of PLGA and chitosan nanospheres were investigated according to the previous study with minor modifications [54]. Loading 2 mg plasmid, the freeze-dried nanospheres were first dissolved in 1 mL PBS (pH 7.4) and then sat on a rotary shaker (180 rpm) at 37 °C. To investigate the release characteristics, the nanosphere solution was first centrifuged at 12,000 rpm for 1 min, and then the total volume of supernatant and plasmid concentration in supernatant were measured. The plasmid concentration in the supernatant was quantified by the nanodrop microvolume spectrophotometer referenced to non-plasmid nanosphere solution under the same procedure. The nanospheres at the bottom were resuspended after measurement, and the interval between two measurements was 12 h. The cumulative release (CR) was analyzed by Formula (3), and the release characteristics were evaluated. Each group had a triple replicate, and each replicate was measured once at one measurement.
(3)$$\mathrm{CR}\;\left(\%\right)=\frac{\mathrm{Total}\;\mathrm{volume}\;\mathrm{of}\;\mathrm{supernatant}\times \;\mathrm{Plasmid}\;\mathrm{concentration}}{\mathrm{Total}\;\mathrm{amount}\;\mathrm{of}\;\mathrm{loaded}\;\mathrm{plasmid}}\times 100\%$$

### 2.8. Animals Immunization Schedule and Challenge

To determine the toxicity of nanospheres on normal animals, 30 BALB/c mice were randomized in six groups. Each mouse was treated intramuscularly with a dose containing 300 μg pVAX1-SIR2, which was three times more than the regular dose for normal patients. The mice in the CS or PLGA group were treated with equal amount of chitosan or PLGA nanosphere loaded with PBS, and those in the blank group were treated with equal volume of PBS. Three days later, a booster immunization with same dose and same immunization method was conducted again. After 24 h of treatment, sera from the eye socket were harvested and the levels of blood urea nitrogen (BUN) were conducted by the commercial kit (Solarbio, Beijing, China). During the test period, the surviving animals were observed every 24 h for mental status and physical health. The mental status mainly included responses of stimulation, while the physical health mainly included diets, abnormal changes in the injection site, and activities.

BALB/c mice were randomly divided into seven groups of 23 mice per group, and the experimental animal received an intramuscular injection containing 100 μg pVAX1-SIR2 plasmids at weeks zero and two. All pVAX1-SIR2 plasmids and synthesized nanospheres were diluted and suspended in PBS (pH 7.4), and all groups were injected with an equal volume of vaccine (each injection did not exceed 0.5 mL). At week four, all animals intraperitoneally received a lethal dose (1 × 10^3^ tachyzoites) injection of *T. gondii* RH strain (Table 1). Described as the predilection tissue of *T. gondii* in mice [55], the cardiac muscle was collected under the supervision of animal ethics committee of responsible authority from the college of veterinary medicine, Nanjing agricultural university, PR China seven days after the challenge. Blood samples were also collected from the eye socket before immunization or challenge at week zero, two, and four.

### 2.9. Determination of Antibodies and Cytokines

According to the previous study [56], indirect enzyme linked immunosorbent (ELISA) was conducted to determine the levels of IgG, IgG1, and IgG2a in sera. Briefly, each well of microtiter plates (96 wells, Costar, Cambridge, MA, USA) was coated with 1 μg rTgSIR2 protein (diluted to 10 μg/mL with carbonate buffer pH 9.6) overnight at 4 °C. After three times washing in TBST (TBS containing 0.05% Tween 20), each well was blocked with TBS containing 5% bovine serum albumin at 37 °C for 1 h. After washing three times in TBST again, the sera of animals were subsequently diluted (1:100 in TBS) and added to each well for 1 h incubation at 37 °C. Rinsed 5 min in TBST, HRP-conjugated anti-mouse IgG, IgG1, or IgG2a (1:5000, eBioscience, San Diego, CA, USA) were added to detect the bound antibodies. As the substrate, 3,3′,5,5′-tetramethylbenzidine (TMB, Tiangen, Beijing, China) was used to reveal the immune complexes. The reaction was stopped by 100 μL 2 M H_2_SO_4_, and a microplate photometer (Thermo Scientific, Waltham, MA, USA) was utilized to detect the absorbance values at 450 nm. Each group had five replications, and each replication was run once.

To detect the levels of cytokines in sera, commercial ELISA kits (Yifeixue, Nanjing, China) were used. According to standard curves obtained from known amounts of mouse recombinant cytokines, the concentrations of interferon-gamma (IFN-γ), interleukin (IL) 4 (IL-4), IL-10, and IL-17 were assessed strictly followed the instructions. Each group had five replications and each replication was investigated once.

### 2.10. Lymphocyte Proliferation and Flow Cytometry

Seven days after the booster immunization (week five), three mice from each group were euthanized to isolate the splenic lymphocytes following the guidelines of lymphocyte separation kit (Solarbio, Beijing, China). The isolated lymphocytes were cultured in a 6-well plate (Costar, Cambridge, MA, USA) at a density of 10^7^ cells/well in DMEM supplemented with 10% (*v/v*) FBS, 100 U/mL penicillin, and 100 mg/mL streptomycin overnight. Then, T cells, B cells, and NK cells were collected, and the suspended cells were transferred into a 96-well plate (10^5^ cells/well) containing 20 μg/mL rTgSIR2 protein. The plate was subsequently incubated in 5% CO_2_ at 37 °C for 72 h, and 10% (*v/v*) reagent from Cell Counting Kit 8 (CCK-8, Beyotime, Shanghai, China) were supplemented according to the manual. Two hours later, the lymphocyte proliferation was assessed by a microplate photometer by measuring the absorbance vales at 450 nm. Each group had three animals, and the spleen lymphocytes from one animal were randomly divided into four replications; each replication was investigated once. Based on Formula (4), the obtained values were described as the stimulation index (SI).
(4)$$\mathrm{SI}\;\left(100\%\right)=\frac{\mathrm{mean}\;A450\;\mathrm{value}\;\mathrm{of}\;\mathrm{the}\;\mathrm{exanimated}\;\mathrm{group}}{\mathrm{mean}\;A450\;\mathrm{value}\;\mathrm{of}\;\mathrm{the}\;\mathrm{blank}\;\mathrm{group}}\times 100\%$$

Before immunization or challenge, five mice from each group were sacrificed at week zero, two, and four, respectively, and the splenic lymphocytes were isolated using a lymphocyte separation kit. For the percentages of major histocompatibility complex (MHC) class I (MHC-I) and II (MHC-II) in dendritic cells (DCs), the lymphocytes were stained with anti-mouse CD11c-PE, MHC-I-FITC, and MHC-II-APC (eBioscience, San Diego, CA, USA). For the CD4^+^ T lymphocyte and CD8^+^ T lymphocyte subsets, the lymphocytes were, respectively, stained with anti-mouse CD3e-PE and CD4-FITC (eBioscience, San Diego, CA, USA), and anti-mouse CD3e-PE and CD8-FITC (eBioscience, San Diego, CA, USA). To analyze the DC maturation, the lymphocytes were cultured overnight as described above, and the adherent cells were obtained. Cells were then stained with anti-mouse CD11c-APC, CD86-FITC, and CD83-PE (eBioscience, San Diego, CA, USA). The staining procedure was carried out in dark for 40 min at 4 °C, and cells were washed three times in PBS (pH 7.4) before flow cytometry analysis (Beckman Coulter Inc, Brea, CA, USA). Before analyzing splenic lymphocytes, the fluorescence compensation was conducted by CytExpert software (version 2.3, Beckman Coulter Inc, Brea, CA, USA) according to the instructions. Each group had five replications and each replication was investigated once.

### 2.11. T. gondii Burdens in Animals

To investigate the parasite burdens in mice, the 30 mg cardiac sample collected in Section 2.8 was used to extract genomic DNA according to the instructions (OMEGA Bio-tek, Norcross, GA, USA), and the extracts were stored at −20 °C until use. To estimate the purity of genomic DNA, the OD260/OD280 value of each sample was conducted by the nanodrop microvolume spectrophotometer before amplification. The recombinant vector, including the amplified region, was also prepared according to the previous study [57], and the copy numbers of recombinant vector were calculated by an online tool (http://cels.uri.edu/gsc/cndna.html, accessed on 28 September 2021). Referencing the published paper with minor modifications [58], absolute quantitative PCR (qPCR) was conducted using the following amplification mixture: 10 μL 2 × ChamQ SYBR qPCR MasterMix (Vazyme, Nanjing, China), 0.4 pmol of each primer, 0.4 μL 50 × ROX Reference Dye 2, 1 μL DNA extracts, and double-distilled water to make t final volume of 20 µL. Amplified by the Applied Biosystems 7500 (Life Technologies, Carlsbad, CA, USA), each reaction was amplified using the following program: 30 s at 95 °C, 40 cycles of 10 s at 95 °C, and 30 s at 60 °C. Before further analysis, the solubility curve of each reaction was ensured with one uniform peak at the expected temperature. Referenced to known amount of recombinant vector, the regression curve was also constructed to calculate the copy numbers in mice. Each group had five replications, and each replication was run in triplicate.

### 2.12. Statistical Analysis

Using GraphPad 8.0 software (GraphPad Prism, San Diego, CA, USA), one-way analysis of variance (ANOVA) was conducted in the levels of antibodies, cytokines, lymphocyte proliferation, flow cytometry analysis, and parasite burdens. Differences between groups were considered as significant at *p* < 0.05, and data were expressed as mean ± standard deviation (SD). Significances between pVAX1-SIR2/CS and pVAX1-SIR2/PLGA group were evaluated with the independent *t*-test.

## 3. Results

### 3.1. Construction of the Recombinant Plasmid and Its Expression

The recombinant plasmid pVAX1-SIR2 was constructed as described above, and the coding sequence of *T. gondii* histone deacetylase SIR2 was 1083 bp and encoded a 361-amino acid protein in theory. To verify the recombinant plasmid, the double digestion was conducted with *Eco*RI and *Xho*I, yielding two fragments with the predicted size of 1095 bp and 2966 bp (Figure 1a). The plasmid sequencing analysis revealed the insert were the open reading frame (ORF) of *T. gondii* SIR2 gene. All these results indicated that the pVAX1-SIR2 plasmid was correctly constructed.

To investigate the expression of pVAX1-SIR2 plasmids in vitro, an immunofluorescence assay was conducted, and cells were stained by serum from rats against rTgSIR2 protein (Figure 1b). HEK 293-T cells transfected with pVAX1-SIR2 plasmids showed specific red fluorescence, whereas cells transfected with blank pVAX1 plasmids did not show red fluorescence. These findings indicated that *T. gondii* SIR2 protein could expressed by pVAX1-SIR2 plasmids in HEK 293-T cells.

### 3.2. Physical Characterization and Release Characteristics

Based on the method of double emulsion evaporation technique (*w/o/w*) and ionic gelation technique, PLGA and chitosan nanospheres loaded with pVAX1-SIR2 plasmids were prepared. As illustrated in Figure 2, the SEM pictures revealed that both PLGA and chitosan nanospheres were spherical in shape and convex in their surfaces. Synthesized by 5% PVA, the average diameter of PLGA nanospheres loaded with pVAX1-SIR2 plasmids reached approximately 91.51 ± 6.84 nm (*n* = 5), while the average diameter of chitosan nanospheres loaded with pVAX1-SIR2 plasmids was around 106.21 ± 17.50 nm (*n* = 5). The EE and LC of pVAX1-SIR2/PLGA nanospheres reached 75.62% and 1.28% (*n* = 3), while those were 54.71% and 3.47% (*n* = 3) in pVAX1-SIR2/CS nanospheres.

Referenced by the PLGA and chitosan nanospheres loaded with PBS, the release characteristics of PLGA and chitosan were investigated. The pVAX1-SIR2/CS nanospheres showed a more favorable release profile (Figure 3), and 18.42 ± 5.80% of the plasmids were detected immediately. The plasmids were slow-released over the following days and finally reached approximately 80%. Unlike pVAX1-SIR2/CS nanospheres, the pVAX1-SIR2/PLGA nanospheres showed a burst release within the first two days and became gentle during the following two days. The release profile was accelerated between day four and day eight, and finally, a trend of stable release was observed.

### 3.3. Toxicity Analysis in Animals

The levels of BUN in sera were investigated to assess the toxicity of nanospheres on mouse’s kidney tissue. Described in Figure 4, the levels of BUN remained in the clinically acceptable normal ranges (*p* > 0.05). According to the clinical observation, all treated animals were stable in mental status compared to the animals immunized with PBS. No adverse reaction was occurred in the treated mice at the injection site, and the diets and activities were similar to those immunized with PBS.

### 3.4. Modulations of Antibodies and Cytokines Secretions in Animals

A standard ELISA procedure was conducted to evaluate the titers of IgG, isotypes IgG1, and IgG2a in mice sera. As depicted in Figure 5a, the levels of IgG in groups pVAX1-SIR2, pVAX1-SIR2/CS, and pVAX1-SIR2/PLGA were significantly higher than those in the blank and control group after the first (week two) and second (week four) immunization. As demonstrated in Figure 5b, animals immunized with pVAX1-SIR2 plasmids, pVAX1-SIR2/CS nanospheres, and pVAX1-SIR2/PLGA nanospheres could generate higher (*p* < 0.001) levels of IgG1 in week two. Compared with the blank or control group, animals in the pVAX1-SIR2, pVAX1-SIR2/CS, and pVAX1-SIR2/PLGA groups could generate significantly higher levels of IgG1 after the booster immunization. As for the IgG2a in Figure 5c, animals immunized with pVAX1-SIR2 plasmids, pVAX1-SIR2/CS nanospheres, or pVAX1-SIR2/PLGA nanospheres could generate higher levels of IgG2a compared with the those in the blank or control group after the first (week two) and second (week four) immunization. In week four, animals immunized with pVAX1-SIR2/CS nanospheres could generate higher levels (*p* < 0.05) of IgG2a than those in pVAX1-SIR2/PLGA group.

Based on double antibody sandwich method, the levels of IFN-γ, IL-4, IL-10, and IL-17 from the animals’ sera were determined by commercial ELISA kits after the second immunization (week four). Compared to the blank or control group (Figure 6a), obviously higher (*p* < 0.001) IFN-γ secretions could be detected in pVAX1-SIR2/CS and pVAX1-SIR2/PLGA group at week two and four, while higher (*p* < 0.001) secretions could be only detected in pVAX1-SIR2 group after the second immunization. In addition, animals in the pVAX1-SIR2/PLGA group could generate higher (*p* < 0.05) levels of IFN-γ than those in the pVAX1-SIR2/CS group. As for the secretions of IL-4 (Figure 6b), significant secretions in animals immunized with pVAX1-SIR2/CS and pVAX1-SIR2/PLGA nanospheres were enhanced after the first (week two) and second (week four) immunization. For the secretion of IL-10 (Figure 6c), significantly higher levels were also observed in animals from pVAX1-SIR2/PLGA group after week two and week four, compared to the blank or control group. As for IL-17 exhibited in Figure 6d, significant secretions could be only observed in pVAX1-SIR2/PLGA group compared with the blank and control group after the first immunization. However, significant secretions could be evaluated in the pVAX1-SIR2 (*p* < 0.05), pVAX1-SIR2/CS (*p* < 0.001), and pVAX1-SIR2/PLGA (*p* < 0.001) groups after the second immunizations, compared with the blank and control groups.

### 3.5. Phenotype Analysis of Dendritic Cells

To identify the targeted cells and evaluate the percentage, CD11c^+^CD83^+^ cells, CD11c^+^CD86^+^ cells, CD11c^+^MHC-I^+^ cells, and CD11c^+^MHC-II^+^ cells in spleen lymphocytes were gated as illustrated in Appendix A. Investigated by flow cytometry, the expression of CD83 and CD86 on DCs were evaluated in Figure 7. Compared to the blank or control group in Figure 7a, the percentage of CD11c^+^CD83^+^ cells from the pVAX1-SIR2, pVAX1-SIR2/CS, and pVAX1-SIR2/PLGA groups was significantly promoted after the first and second immunization (week two and week four). No statistically difference (*p* > 0.05) was revealed between the pVAX1-SIR2/CS and pVAX1-SIR2/PLGA groups in CD11c^+^CD83^+^ expression. As for CD11c^+^CD86^+^ expression (Figure 7b), treatment with pVAX1-SIR2/CS and pVAX1-SIR2/PLGA nanospheres showed a significant improvement in expressing CD11c^+^CD86^+^ when compared with the blank and control group after the first immunization (week two). Furthermore, a significant improvement was also revealed in animals immunized with pVAX1-SIR2 plasmids (*p* < 0.01), pVAX1-SIR2/CS nanospheres (*p* < 0.001), and pVAX1-SIR2/PLGA nanospheres (*p* < 0.001) after the second immunization (week four). Noticeably, animals from pVAX1-SIR2/CS group could express higher levels (*p* < 0.001) of CD11c^+^CD86^+^ than those from the pVAX1-SIR2/PLGA group.

As the surface maker of splenic DCs, the expression of MHC molecules was also investigated by flow cytometry. Compared with the blank or control group (Figure 8a), the relative percentage of CD11c^+^MHC-I^+^ cells were obviously (*p* < 0.01) promoted by pVAX1-SIR2/CS and pVAX1-SIR2/PLGA nanospheres after the second immunization (week four), and animals generated equal percentage (*p* > 0.05) of MHC-I molecules after the first immunization (week two). However, as shown in Figure 8b, the ratio of MHC-II molecules in animals from pVAX1-SIR2, pVAX1-SIR2/CS, and pVAX1-SIR2/PLGA group were significantly (*p* < 0.001) enhanced after the first (week two) and second (week four) immunization. Based on the independent *t*-test, MHC-II molecules in pVAX1-SIR2/PLGA group were remarkably (*p* < 0.001) promoted when compared to those in pVAX1-SIR2/CS group after the first immunization (week two).

### 3.6. Lymphocyte Proliferation

One week after the booster immunization (week five), animals were sacrificed and spleen lymphocytes were collected, then the proliferative responses were analyzed. Demonstrated in Figure 9, animals from pVAX1-SIR2, pVAX1-SIR2/CS, or pVAX1-SIR2/PLGA group could generate a significant (*p* < 0.001) stimulation index when compared with the blank and control group. In addition, animals immunized with pVAX1-SIR2/CS nanospheres could generate a promoted proliferation (*p* < 0.01) compared with those immunized with pVAX1-SIR2/PLGA nanospheres.

### 3.7. Proportions of T Lymphocytes

In order to analyze the percentages of CD4^+^ and CD8^+^ T cells in splenic T lymphocytes, five mice from each group were euthanized, and the splenic lymphocytes were conducted by flow cytometry. To identify the targeted cells and evaluate the percentage, CD3e^+^CD4^+^ and CD3e^+^CD8^+^ cells in spleen lymphocytes were gated as illustrated in Appendix A. Derived from pVAX1-SIR2/CS and pVAX1-SIR2/PLGA group (Figure 10a), CD3e^+^CD4^+^ cells were significantly promoted after the first immunization (week two) when compared with the blank or control group. After the second immunization (week four), the generation of CD3e^+^CD4^+^ cells were promoted in animals from pVAX1-SIR2, pVAX1-SIR2/CS, and pVAX1-SIR2/PLGA group. In case of the percentage of CD8^+^ T lymphocytes (Figure 10b), statistically higher levels of CD3e^+^CD8^+^ cells could be detected in animals immunized with pVAX1-SIR2 plasmids (*p* < 0.05), pVAX1-SIR2/CS nanospheres (*p* < 0.001), and pVAX1-SIR2/PLGA nanospheres (*p* < 0.001) at week four, while enhanced generations of CD3e^+^CD8^+^ cells could be observed in pVAX1-SIR2/CS groups at week two.

### 3.8. T. gondii Burdens in Animals

By detecting the copy number of 529 bp repeat element in cardiac muscles, absolute qPCR was carried out to obtain a more accurate analysis of *T. gondii* burdens in animals. Before amplification, the nanodrop microvolume spectrophotometer was used for evaluating the OD260/OD280 value of each sample, and all samples were in the range of 1.6–1.8. For the *T. gondii* burdens in Figure 11, a remarkable decrease (*p* < 0.001) was observed in animals immunized with pVAX1-SIR2 plasmids, pVAX1-SIR2/CS nanospheres, and pVAX1-SIR2/PLGA nanospheres when compared to those from the blank and control groups.

## 4. Discussion

In our present study, the pVAX1-SIR2 plasmids were entrapped in PLGA and chitosan nanospheres. The results suggested that PLGA and chitosan nanospheres were effective at DNA vaccine delivery and anti-plasmid biodegradation, and they were safe to the animals. In addition, activated DCs and T lymphocytes, evoked humoral and cellular immune responses, and lower parasite burdens were observed; all these findings were evidence that mice immunized with pVAX1-SIR2/CS or pVAX1-SIR2/PLGA nanospheres could be a promising approach to prevent highly virulent *T. gondii* RH strain.

To improve the protective efficacy, nanotechnology has been widely used as vehicles to deliver vaccines [42,59], and many methods have been developed for nanosphere synthesis [49,50]. Different methods of nanosphere formulation can affect physical characteristics, and in the present study, the modified ionic gelation technique and double emulsion solvent evaporation technique were conducted for chitosan and PLGA nanosphere formulation. With 2.5 times higher than 1000 nm-sized nanospheres in absorptance, the 100 nm-sized nanospheres were easily absorbed in HeLa cell lines [60]. The average diameters of chitosan and PLGA nanospheres were 106.21 ± 17.50 nm and 91.51 ± 6.84 nm, respectively, leading to a better entrance into cells. Furthermore, EE and LC are important values in assessing the nanospheres. Following a similar strategy, PLGA nanospheres whose EE reached 57.5% were formulated by Leya et al. [61], while the EE and LC of chitosan nanospheres reached 92.8% and 63.7% in the published report [62]. Such differences may be related to different antigens or produce, and further studies should optimize the procedures and enhance the immune efficiency of nanospheres.

In the present study, both chitosan and PLGA nanospheres were in round shape with many convex structures on the surface. Such properties ensured no obvious degradation of antigens during loading and release processes, and the release curves further confirmed. The PLGA nanospheres showed a triphasic release curve, and such a profile may be affected by the size, porosity, molecular weight, or even type of antigens [63]. Compared with the PLGA nanospheres, chitosan nanospheres showed a steadier release curve without burst release. As a cationic polymer negatively charged, chitosan nanospheres can bind to the surface of cells, resulting in a constant residence in targeted cells and a slow release of antigens [64]. Noticeably, a steady release profile does not represent its applications in vivo, and the toxicity cannot be ignored. DCM used in formulation of PLGA nanospheres were considered as toxic and hard to remove by evaporation [65]. Thus, both chitosan and PLGA nanospheres were completely freeze-dried. As expected, the levels of BUN were similar, and no adverse reaction occurred, indicating the nanospheres were nontoxic.

In vaccination studies, the vaccine strategy is critical in successful immunization against *T. gondii*. In the related literatures, various vaccine strategies were conducted, and no study explained why a certain program was used. Understanding the durability of vaccine-induced immunity is important for making advisable decisions on the recommended time interval between booster vaccinations. In the present study, a two-week interval was selected based on the published literature related to the vaccines for *T. gondii* [56,66], and the optimal time interval for the formulated nanospheres should be determined in future studies. Compared with the published paper, the DNA vaccine encoding SAG1, SAG3, and SAG5 with CpG-ODN adjuvant could elicit significant production of IgG, IgG1, and IgG2a after the last immunization [67]. Moreover, significantly higher levels of IgG, IgG1, and IgG2a could be also revealed by the *T. gondii* DNA vaccine, pVAX1-MYR1 [56]. Many constructed *T. gondii* vaccines have reported the similar results in eliciting high levels of IgG, IgG1, and IgG2a after the last immunization, such as the multi-epitope ROP8 DNA vaccine [68], DNA vaccine encoding *T. gondii* ROP35 [69]. Due to different vaccine strategies, the check point for antibodies and cytokines varied from one study to another. Such differences made the comparison between two vaccines more difficult. Thus, a standardized operation should be formed in the subsequently researches for large-scale screening of promising vaccines against *T. gondii*.

Th1-related immunity can be induced and generate specific IgG, which can decrease the adhesion rate of *T. gondii* to targeted cells and then limit its replication [70]. Largely secreted IgG plays an important role in resisting the infection of *T. gondii* and inhabiting the reactivation of cysts [71]. Due to IgG1 and IgG2a being associated with Th2-related and Th1-related immunity [72], the titers of IgG2a antibodies was higher than that of IgG1 in week two, indicating a Th1-dominated immunity generated. However, in week four, a mixed Th1/Th2 immune response was developed with the changes of IgG1 and IgG2a secretions. These findings lent credit to the idea that types of nanospheres could finally generate a mixed Th1/Th2 immune response, and such immune type were similar to the pVAX1-SIR2 immunization. Similar to the present results, both cell (Th1-related) and humoral (Th2-related) mediated immunity could be induced by pVAX1-MYR1 in BABL/c mice [56]. Furthermore, with the promoted titers of IgG, IgG1, and IgG2a in vaccinated mice, our results also indicated that a booster immunity could be induced by chitosan or PLGA.

Cytokines are crucial in modulating Th cells, and increased secretion of Th1 cytokines is a key mechanism for resisting *T. gondii* infections [73]. During nature *T. gondii* invasion, high levels of IFN-γ can activate phagocytes and determine the fate of *T. gondii* [74,75]. In the present study, increased secretions of IFN-γ were evaluated in vaccinated animals, leading to a Th1-related immunity. In addition, high levels of IL-4 were also detected, which are associated with Th2-related immunity. It has been demonstrated that early mortality of animals infected with acute toxoplasmosis is induced by severe systemic inflammation [76]; thus, large secretions are also critical. According to the published paper, IL-4 can enhance the secretions of IFN-γ in the late stage of *T. gondii* infections [77]. Associated with increased pro-inflammatory cytokines during toxoplasmosis, IL-10 plays a central role in reducing inflammation and severe immunopathology mediated by CD4^+^ T cells [78]. Promoted secretions of IL-10 were only obtained in animals from pVAX1-SIR2/PLGA group, and such a difference may be linked to excessive humoral immunity. Produced by Th17 cells, cytokine IL-17 are also involved in anti-*T. gondii* infections [79,80], and functioned as a tissue inflammatory regulator [81]. In the current study, all vaccinated animals were detected significant high levels of IL-17 after the second immunization. This finding indicated that pVAX1-SIR2 plasmids and its nanospheres could induce the differentiation of Th17 cells and participate in the resistance of *T. gondii*.

As the potent professional antigen present cell (APC), DCs play a critical role in innate and adaptive immunity [82]. After capturing antigens, DCs become matured and start presenting antigens, and then activate naïve T cells [83,84]. Immature DCs cannot effectively activate T cells, but the capacity boosts if DCs obtain a maturation stimulus [85]. When a maturation stimulus received, surface molecules are expressed on the surface of DCs, which play important role in activation of T cells [86]. CD86 is essential for T cells activation and adaptive immunity development [87], and CD83 mainly expressed on the surface of mature DCs, which was an important checkpoint for immunology [88]. In the current study, all vaccinated groups had shown obvious increasing tendencies in CD11c^+^CD83^+^ and CD11c^+^CD86^+^ cells, indicating that pVAX1-SIR2/CS and pVAX1-SIR2/PLGA nanospheres could strongly promote the DC maturation and generate its functions. To carry out the function of antigen presentation, the matured DCs have developed MHC class II molecules, which could present antigens [89]. Localized on the surface of mature DCs, the MHC-II molecules conjugated with antigens can be easily recognized by CD4^+^ T lymphocytes [90], finally leading to a strong immune response [91,92]. Expressed in every nucleated cell, MHC-I molecules play an essential role in endogenous antigens presentation, finally activating CD8^+^ T lymphocytes [93]. Through a process named cross-presentation, DCs are able to present antigens from extracellular sources on their MHC-I molecules [94]. Promoted by pVAX1-SIR2/CS and pVAX1-SIR2/PLGA nanospheres, DCs could produce massive MHC-II molecules as well as MHC-I molecules on its surface. A previous paper demonstrated that the enhanced expression of MHC-I molecules could be induced by IFN-γ through the Janus kinase/signal transducer and activator of transcription (JAK/STAT) pathway [95]. Thus, the promoted expressions of MHC-I molecules may be related to the high-rising IFN-γ. These findings suggested that both pVAX1-SIR2/CS and pVAX1-SIR2/PLGA nanospheres could induce the exogenous (MHC-II dependent) and endogenous (MHC-I dependent) antigen presentation, and the exogenous antigen presentation were more frequently.

Stimulated by mature DCs, the activated T lymphocytes will experience two major changes: proliferation and differentiation [96]. The proliferation of T lymphocytes was considered the most important value in evaluating immunity status [97]. Based on the current data, the proliferation abilities of splenic lymphocytes from pVAX1-SIR2, pVAX1-SIR2/CS, and pVAX1-SIR2/PLGA group were obviously enhanced, especially from the pVAX1-SIR2/CS group. These findings suggested that both pVAX1-SIR2/CS and pVAX1-SIR2/PLGA nanospheres could largely promote the proliferation of splenic lymphocytes, and the chitosan delivery system was more efficient. The differentiation of T lymphocytes determine the Th1/Th2 bias [98]. Furthermore, CD4^+^ T lymphocytes are important in developments of memory CD8^+^ T lymphocytes after vaccinations [99]. In the present study, the percentages of CD4^+^ and CD8^+^ T lymphocytes were obviously promoted in all immunized animals. These findings made it clear to the idea that pVAX1-SIR2/CS and pVAX1-SIR2/PLGA nanospheres were important in formulations of CD4^+^ and CD8^+^ T lymphocytes, which may be critical in mediating cellular and humoral immunity against *T. gondii*.

By being challenged with the virulent RH strain of *T. gondii*, the parasite loads were investigated to evaluate the immune protection. In comparison with the controls, animals from the pVAX1-SIR2, pVAX1-SIR2/CS, and pVAX1-SIR2/PLGA groups presented a significant inhibition of *T. gondii* replications, indicating a good immune protection against *T. gondii*. At present, the vaccine that could provide completely immune protection against acute *T. gondii* infections is still unavailable [15,17]. Thus, the pVAX1-SIR2/CS and pVAX1-SIR2/PLGA nanospheres may have more advantages for inhibition of acute *T. gondii* replications.

## 5. Conclusions

Humoral and cellular immune responses could be induced by pVAX1-SIR2/CS and pVAX1-SIR2/PLGA nanospheres immunization, which were a main reason leading to a protection against *T. gondii*. Co-administration of chitosan and PLGA could promote the immune protection and efficacy of DNA vaccines, and the two types of vaccines were also revealed to be effective in reducing parasite burdens in cardiac muscles. Furthermore, as substitutes for each other, pVAX1-SIR2/CS and pVAX1-SIR2/PLGA nanospheres can induce similar immunity and might be the promising vaccines for further investigations. In further investigations, the loading capacity should be improved and the potency of the pVAX1-SIR2/CS and pVAX1-SIR2/PLGA nanospheres should be validated using different strategies, such as different animal models, different *T. gondii* strains, and chronic infections.

## Figures and Tables

**Figure 1 pharmaceutics-13-01582-f001:**
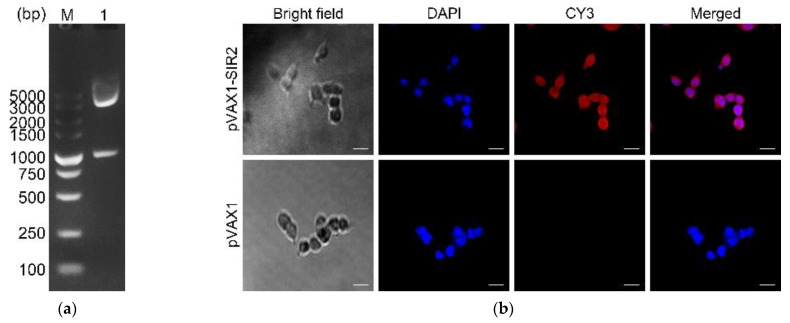
(**a**) Double digestion analysis of the recombinant plasmid pVAX1-SIR2. Line M: DL5000 marker; Line 1: Double digestion with *Eco*RI and *Xho*I. (**b**) Fluorescent microscopy imagines of HEK 293-T cells transfected with pVAX1-SIR2 and pVAX1 plasmids. Cells were detected by sera from rats against rTgSIR2 protein. Bar represents 25 μm.

**Figure 2 pharmaceutics-13-01582-f002:**
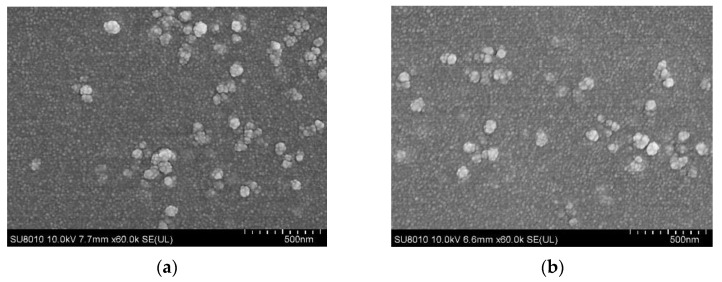
SEM images of PLGA (**a**) and chitosan nanospheres (**b**) loaded with pVAX1-SIR2 plasmids. PLGA and chitosan nanospheres were synthesized by double emulsion evaporation technique (*w/o/w*) and ionic gelation technique. After being totally freeze-dried, nanospheres were imagined at magnification of ×30,000 (bar represented 500 nm).

**Figure 3 pharmaceutics-13-01582-f003:**
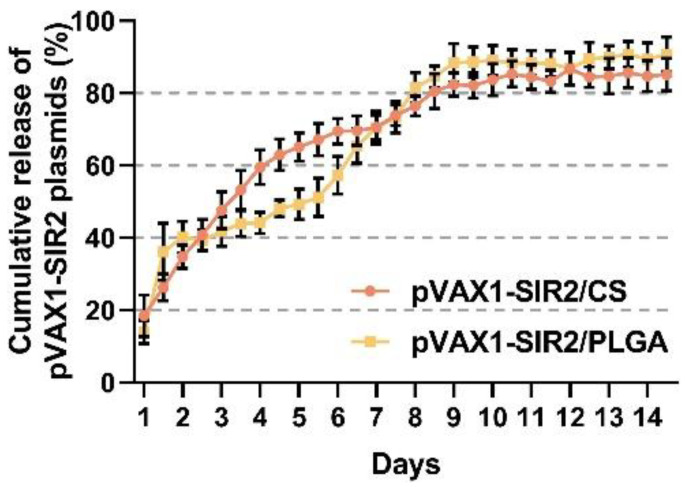
The release characteristics of PLGA and chitosan nanosphere loaded with pVAX1-SIR2 plasmids in vitro. The amount of free pVAX1-SIR2 plasmids in the supernatant was measured by the nanodrop microvolume spectrophotometer. Each group had three replications and each replication was investigated once. Values were represented as mean ± SD (*n* = 3).

**Figure 4 pharmaceutics-13-01582-f004:**
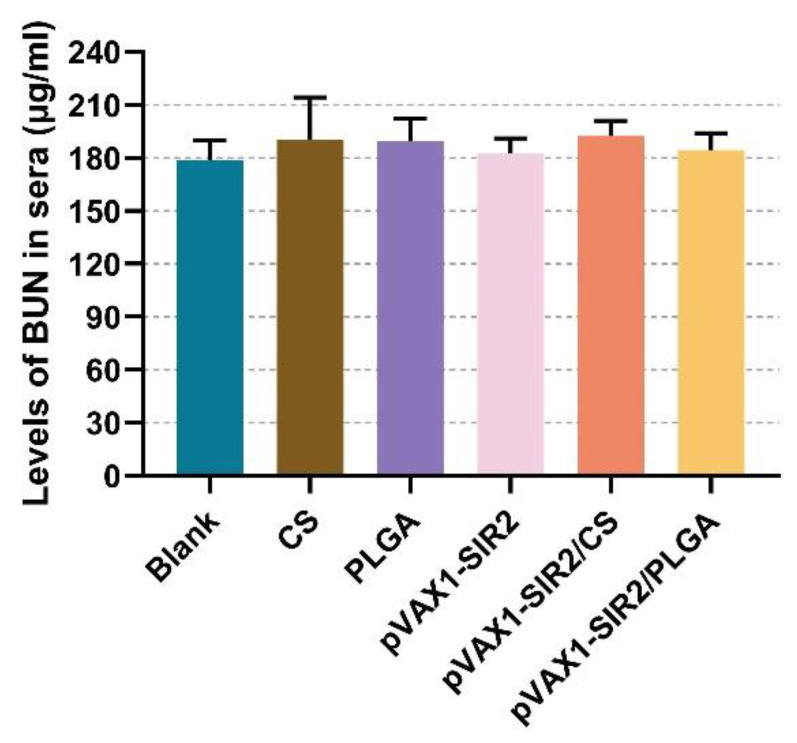
Toxicity analysis of PLGA and chitosan nanospheres loaded with pVAX1-SIR2 plasmids. Five serum samples from each group were separated and the levels of BUN were investigated by the spectrophotometry immediately after blood collection. Each replication was investigated once, and values were analyzed by one-way ANOVA analysis followed by Dunnett’s test. Values were shown as mean ± SD (*n* = 5).

**Figure 5 pharmaceutics-13-01582-f005:**
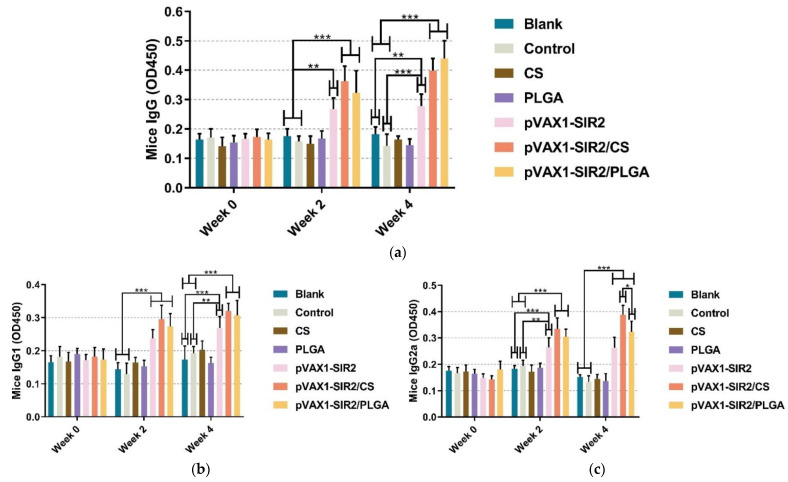
Secretions of IgG (**a**), isotypes IgG1 (**b**), and IgG2a (**c**) in the sera separated from immunized animals at week zero, two, and four. The dilution ratio of tested sera was 1:100 in ELISA analysis, and each serum was investigated once. Values were analyzed by one-way ANOVA analysis followed by Dunnett’s test. Values between the pVAX1-SIR2/CS and pVAX1-SIR2/PLGA group were estimated by the independent *t*-test. Values were shown as mean of OD450 ± SD (*n* = 5). * *p* < 0.05, ** *p* < 0.01, and *** *p* < 0.001 compared with blank or control group.

**Figure 6 pharmaceutics-13-01582-f006:**
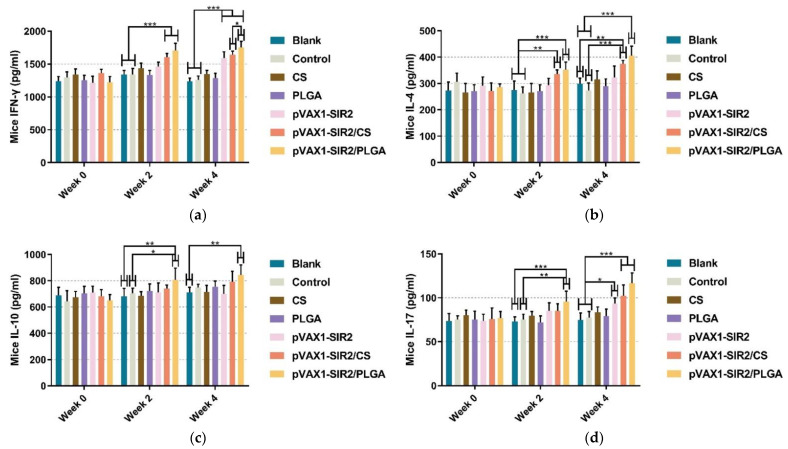
Cytokine secretions in animals. Using the commercial ELISA kits, the secretions of IFN-γ (**a**), IL-4 (**b**), IL-10 (**c**), and IL-17 (**d**) in animals’ sera were investigated at week zero, two, and four. Each serum was investigated once, and values were analyzed by one-way ANOVA analysis followed by Dunnett’s test. Values between the pVAX1-SIR2/CS and pVAX1-SIR2/PLGA group were estimated by the independent *t*-test. Values were shown as mean ± SD (*n* = 5). * *p* < 0.05, ** *p* < 0.01, and *** *p* < 0.001 compared with blank or control group.

**Figure 7 pharmaceutics-13-01582-f007:**
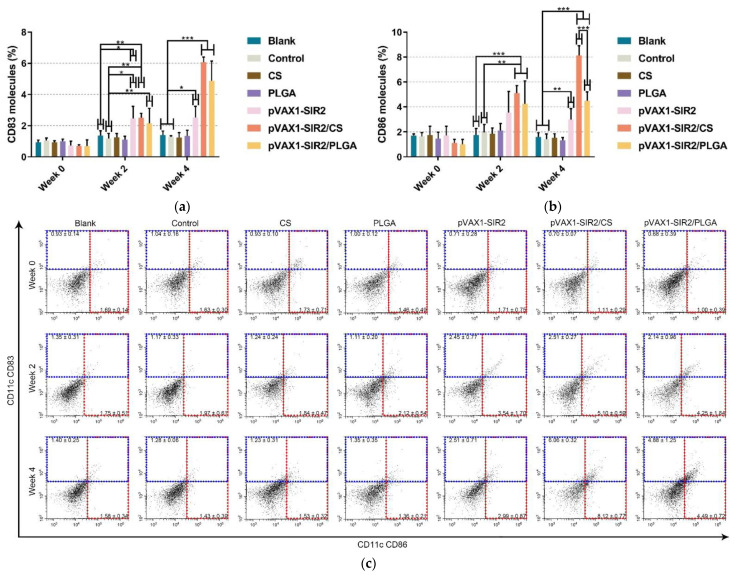
Analysis of splenic DC maturation at week zero, two, and four. The expression of CD83 and CD86 molecules on DCs was evaluated by flow cytometry. Bar graph showed the ratio of CD83 (**a**) and CD86 (**b**) molecules on the surface of splenic DCs, and the dot plots (**c**) showed the percentage of CD11c^+^ CD83^+^ and CD11c^+^ CD86^+^ cells. Each sample was investigated once, and values were analyzed by one-way ANOVA analysis followed by Dunnett’s test. Values between the pVAX1-SIR2/CS and pVAX1-SIR2/PLGA group were estimated by the independent *t*-test. Values were shown as mean ± SD (*n* = 5). * *p* < 0.05, ** *p* < 0.01, and *** *p* < 0.001 compared with blank or control group.

**Figure 8 pharmaceutics-13-01582-f008:**
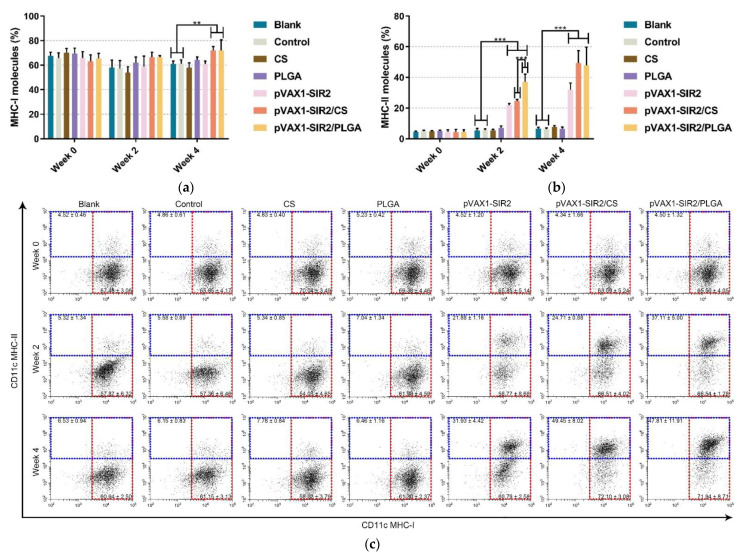
Analysis of MHC molecules on the surface of DCs at week zero, two, and four. The expression of MHC-I and MHC-II molecules was evaluated by flow cytometry. Bar graph showed the ratio of MHC-I (**a**) and MHC-II (**b**) molecules on splenic DCs, and the dot plots (**c**) showed the percentage of CD11c^+^MHC-I^+^ and CD11c^+^MHC-II^+^ cells. Each sample was investigated once, and values were analyzed by one-way ANOVA analysis followed by Dunnett’s test. Values between the pVAX1-SIR2/CS and pVAX1-SIR2/PLGA group were estimated by the independent *t*-test. Values were shown as mean ± SD (*n* = 5). ** *p* < 0.01 and *** *p* < 0.001 compared with blank or control group.

**Figure 9 pharmaceutics-13-01582-f009:**
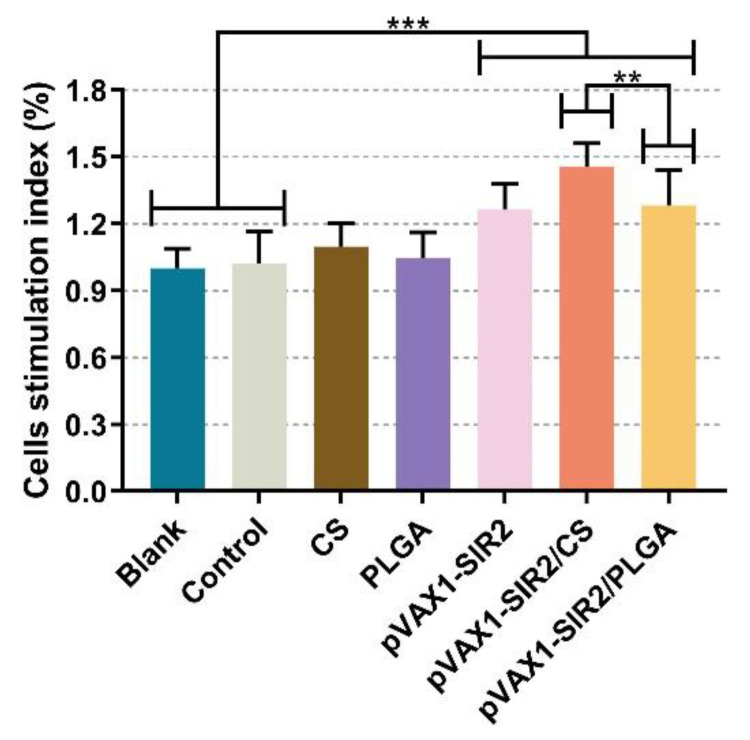
Splenocyte proliferation of immunized animals. Three mice from each group were euthanized, the spleen lymphocytes were collected, and the lymphocytes from one mouse were randomized into four replications. Then, the lymphocytes were cultured with 20 μg/mL rTgSIR2 protein. Each replication was investigated once, and values were analyzed by one-way ANOVA analysis followed by Dunnett’s test. Values between the pVAX1-SIR2/CS and pVAX1-SIR2/PLGA group were estimated by the independent *t*-test. Values were shown as mean ± SD (*n* = 3). ** *p* < 0.01 and *** *p* < 0.001 compared with blank or control group.

**Figure 10 pharmaceutics-13-01582-f010:**
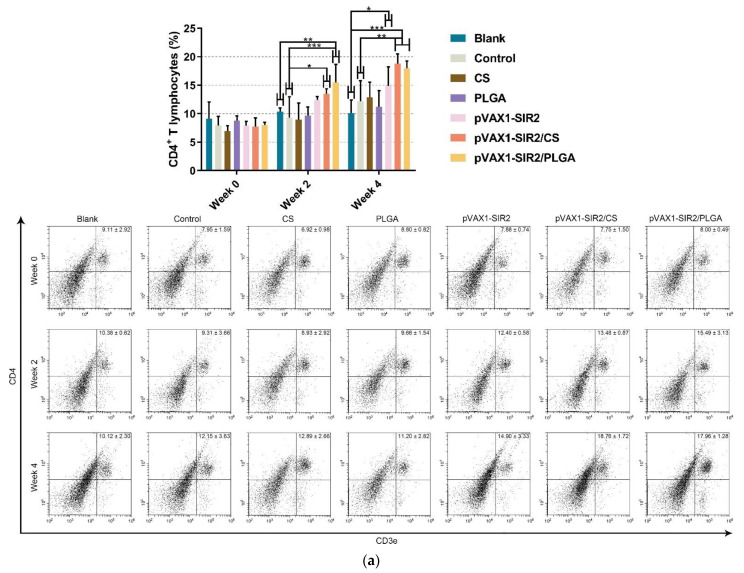
Proportions of CD4^+^ and CD8^+^ T lymphocytes at week zero, two, and four. Stained with CD3e-PE, CD4-FITC (**a**) or CD3e-PE, CD8-FITC (**b**), the separated lymphocytes were evaluated by flow cytometry. Each sample was investigated once, and values were analyzed by one-way ANOVA analysis followed by Dunnett’s test. Values between the pVAX1-SIR2/CS and pVAX1-SIR2/PLGA group were estimated by the independent *t*-test. Values were shown as mean ± SD (*n* = 5). * *p* < 0.05, ** *p* < 0.01, and *** *p* < 0.001 compared with blank or control group.

**Figure 11 pharmaceutics-13-01582-f011:**
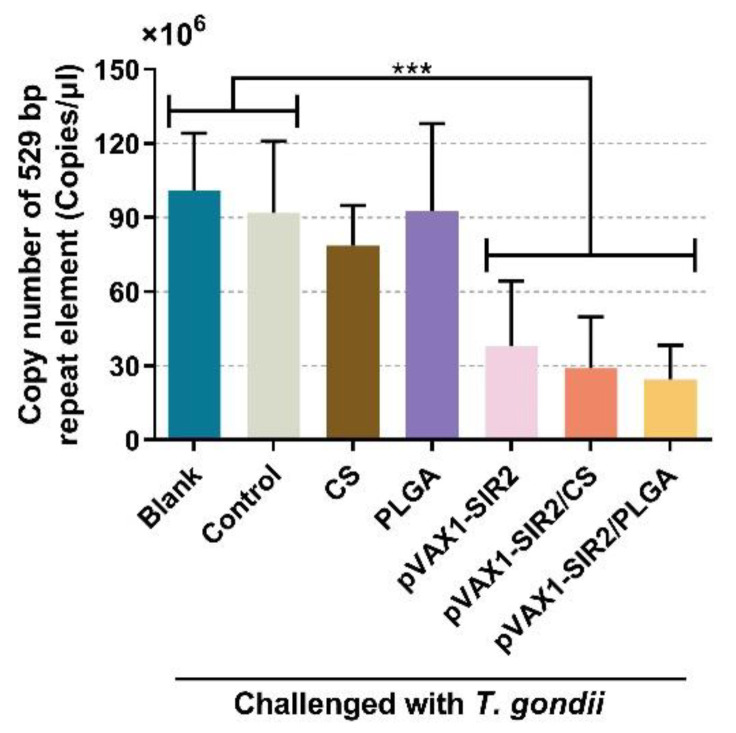
Copy number of *T. gondii* 529 bp repeat element in cardiac muscles. Immunized animals were challenged with 10^3^ tachyzoites two weeks after the second immunization (week four). Seven days after the challenge, cardiac muscles were harvested and investigated. Each sample was investigated three times and values were analyzed by one-way ANOVA analysis followed by Dunnett’s test. Values between the pVAX1-SIR2/CS and pVAX1-SIR2/PLGA group were estimated by the independent *t*-test. Values were shown as mean ± SD (*n* = 5). *** *p* < 0.001 compared with blank or control group.

**Table 1 pharmaceutics-13-01582-t001:** Immunization and challenge procedures in mice.

Group	Treatment (Each Mouse)	Time for Immunization	Time for Infection	Infection Dose (Each Mouse)
Blank	Equal volume of PBS	Week zero and two	Week four	1 × 10^3^ *T. gondii* for each mouse
Control	100 μg pVAX1 plasmids			
CS	Equal amount of chitosan nanospheres loading PBS			
PLGA	Equal amount of PLGA nanospheres loading PBS			
pVAX1-SIR2	100 μg pVAX1-SIR2 plasmids			
pVAX1-SIR2/CS	PLGA nanospheres loading 100 μg pVAX1-SIR2 plasmids			
pVAX1-SIR2/PLGA	Chitosan nanospheres loading 100 μg pVAX1-SIR2 plasmids			

## Data Availability

The data presented in this study are available within the article.

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
