# Peer review of "Nano DNA Vaccine Encoding Toxoplasma gondii Histone Deacetylase SIR2 Enhanced Protective Immunity in Mice"

_pharmaceutics, 2021, doi:10.3390/pharmaceutics13101582_

Round 1
Reviewer 1 Report
I read with interest the manuscript entitled "Nano DNA vaccine encoding Toxoplasma gondii histone 2 deacetylase SIR2 enhanced protective immunity in mice" by Yu et al. In this paper, the authors generated and characterized a DNA vaccine encoding the T. gondii histone 2 deacetylase SIR2 as an antigen and chitosan or PLGA as vaccine carriers (nanosphere). DNA vaccine alone or encapsulated into the nanospheres were used to immunize mice and immune responses and protection after challenge with a lethal dose of T. gondii RH strain were evaluated.
Although the rational and the aim of the study are clearly defined and described, several important aspects need to be better addressed to support the conclusions.
The carrier control groups (chitosan or PLGA without the plasmid) are missing in the in vivo experiments. This is an important issue because immune responses can be activate/modulated by the carrier per se, without increasing the antigen-specific immunity. A comparison with a gold standard vaccine should have been performed or at least discussed. Overall, the immunological data are not convincing, either because of the methods used to evaluate the immune responses or because of the very low immunity induced by the vaccine candidates. In addition, the durability of the vaccine-induced immunity has not been evaluated.
Antibody response: The OD values are useful for the initial screening of sera, but a quantification should be performed to demonstrate the efficiency of a vaccine candidate. The endpoint titer and/or antibody concentration should be evaluated to assess the antibody response. Based on the description of the ELISA in Materials and Methods, total IgG means anti-SIR2 specific IgG. The term “total” here is used for define all isotypes, but is misleading, since total IgG include all the IgG present in serum, regardless of the specificity. Please clarify and modify in the text and figures. Also, as shown in figure 5, the OD values of IgG in blank and control mice are unusually high, indicating a high non specific binding in the assay. Finally, in the figure 5 legend as well as in results, it should be specify that the serum dilution tested in the ELISAs was 1:100.
The levels of antibodies (measured as OD) did not significantly increase after the second immunization, indicating a lack of boost effect. The antibody response is generally very low, considering other vaccines. Please discuss the results comparing them with other vaccine candidates. Is there any known correlate of protection against T.gondii challenge? If so, please discuss better the model and compare with known effective vaccines (live attenuated?).
Cytokines in sera: By looking at the data (figure 6), although the stat analysis indicated significantly differences in cytokine secretion among groups, it seems that the effect of the different immunizations in the modulation of serum cytokines compared to control mice is minor, less than 2-fold increase. Does these levels affect the vaccine efficacy? Please discuss.
Lymphocyte proliferation: the method used for assessing the T cell proliferation is not optimal. The CCK-8 method is more appropriate for evaluating cytotoxicity of drugs or other compounds. Another method should be used (i.e fluorescence dyes as CFSE). Also, a positive control of lymphocyte proliferation, such as ConA, must be included and presented in figure 9. The OD values (very low) compared to controls are not enough informative. Is there another way to present the results other than the OD values? A standard curve that allows a more accurate quantification?
Activation of lymphocytes: The efficacy of a vaccine is usually evaluated as the ability to induce an antigen-specific immunity. In particular, ELISpot and intracellular staining for cytokines following antigen-specific stimulation are the standard methods for assessing the presence and the quality of vaccine-specific T cells. The authors showed only a phenotypic analysis of CD4+ and CD8+ T cells after immunizations. The analysis of T cell activation must be performed whitin the antigen-specific T cell population. Indeed, the general activation of T cells is not a marker of an efficient vaccine, on the contrary could be detrimental. The flow cytometry analysis should be modified since the cell populations (i.e. CD3+CD8+) are not well separated. The gate strategy of the analysis should be shown.
Challenge experiment: although the vaccine candidates showed a significant reduction of copies in cardiac muscle, a survival curve could give more information about the vaccine efficacy.
Minor points:
Lane 642-644: wrong sentence about the origin of Th1 and Th2 cells, please check carefully
Chapter 2.4 should be split in two sections: “generation of anti SIR2 polyclonal antibodies” and “immunofluorescence staining”
Fig 4: specify the timing of analysis
Lane 308: “supernatant” is not the correct term. Delete “the supernatants which contains” and rephrase as: Then T cells, B cells and NK cells were collected…
Lane 494: the sentence …”secrete an enhanced proliferation” is not correct, please modify.

Reviewer 2 Report
Dear Authors,
The article “Nano DNA vaccine encoding Toxoplasma gondii histone deacetylase SIR2 enhanced protective immunity in mice” is interesting and it is written in a correct manner reporting promising findings based on properly selected experiments. The article structure and the discussion are appropriate. Moreover, the manuscript describes in a detailed manner the influence of DNA vaccine encoding T. gondii histone deacetylase SIR2 (pVAX1-SIR2) on the BALB/c mice model (2 vaccine variants: chitosan or PLGA nanospheres loaded with the DNA vaccine) and highlights their immunostimulating properties (both humoral and cellular responses). However, Authors should address several critical concerns to improve the quality of the paper, from a scientific point of view. Summing up, the manuscript requires some minor changes before acceptance, which are listed below. Furthermore, the Authors should also check for possible grammar or spelling typos.
Line 192: it should be „Coverslips” instead „Cover lips”.
Please check the grammar of the article, e.g. line 206.
Fig. 1 please enlarge the microscope images, because they are too small.
Fig. 5 please provide larger graphs.
Fig. 7 Please enlarge the graphs (maybe it is worth placing charts one above the other?).
Fig.8 enlarge as the figure is illegible.
Flow cytometry images should be enlarged as their quality is too poor.
The conclusions could be broadened a bit.
Kind regards
Reviewer 3 Report
In this paper, pVAX1-SIR2/Cs and pVAX1-SIR2/PLGA nanospheres show vaccine efficacy by delivering genes well. pVAX1-SIR2 itself has lower immune efficacy than nanospheres, but seems to be effective for T. gondii burden in cariac muscles.
Therefore, there is a disappointment that it would have been better if pVAX1-SIR2 was compared with the case of electric shock or subcutaneous injection rather than just intramuscular injection.
In addition, if group 2 was treated with nanospheres rather than vector only, the effect of CS or PLGA on T cell immunity could be excluded, and a clear conclusion could be drawn.
It is recommended to mention these parts in the discussion.
Round 2
Reviewer 1 Report
Although the authors modified the manuscript following the requests of the referees, they did not address properly some issues/questions regarding the evaluation of antigen-specific responses. Antibody levels, cytokine expression and lymphocyte proliferation are not impressive, considering the very low increase compared to the control groups. A statistically significant difference between groups does not imply a biological significance or relevance. The ability to control the parasite growth after challenge might be correlated to other vaccine-induced responses, including the innate response, as evaluated by the DC activation and the antigen-specific T cell response, not measured in this paper. These aspects should be considered and discussed, avoiding overstatement as in the Conclusion section (lane 668: “high humoral and cellular response could be induced…”).
Please comment on the following aspects and if possible perform new experiments to address the issues:
Antibody response: The OD values are useful for the initial screening of sera, but a quantification should be performed to demonstrate the efficiency of a vaccine candidate. The endpoint titer and/or antibody concentration should be evaluated to assess the antibody response. Also, as shown in figure 5, the OD values of IgG in blank and control mice are unusually high, indicating a high non specific binding in the assay.
Activation of lymphocytes: The efficacy of a vaccine is usually evaluated as the ability to induce an antigen-specific immunity. In particular, ELISpot and intracellular staining for cytokines following antigen-specific stimulation are the standard methods for assessing the presence and the quality of vaccine-specific T cells. The authors showed only a phenotypic analysis of CD4+ and CD8+ T cells after immunizations. The analysis of T cell activation must be performed whitin the antigen-specific T cell population. Indeed, the general activation of T cells is not a marker of an efficient vaccine, on the contrary could be detrimental.
The wrong sentence about the origin of Th1 and Th2 cells is still there (lane 670-673). “Th1 cells (originated from CD4+ T lymphocytes)”…. and: “Th2 cells (originated from CD8+ T lymphocytes)”. As specified in the cited ref 95, CD4 T cells may differentiate into one of several lineages of T helper (Th) cells, including Th1, Th2, Th17, and iTreg, as defined by their pattern of cytokine production and function. Therefore, it is not clear what the sentence means. In addition, as per ref 97, the protective correlate in T.gondii challenge model is a Th1-biased CD8+ T cells. Please clarify.
Lane 591-602. “Administration program” should be replaced with “vaccine strategy” or “vaccine approach”. Even if the strategies used in different studies are different, the level of immune response at the peak after the last immunization can be compared.
The gate strategy of flow cytometry analysis shown in figures 7 and 11 should be moved to supplementary material.
Round 3
Reviewer 1 Report
I appreciated the modifications the authors have made to improve the manuscript.
But I would like to underline that in the vaccine field the analysis of vaccine-induced immune responses is fundamental. This is not my opinion; it is a fact. Please review the literature and you will find a tremendous number of papers describing the induction of specific antibodies (expressed as endpoint titer or concentration) and of antigen-specific T cell responses after vaccination in preclinical and clinical studies. See these 2 very recent papers as examples of anti-SARS-CoV-2 vaccine (https://www.nature.com/articles/s41541-020-00243-x; https://doi.org/10.1016/j.chom.2020.12.010)
The general activation of T cells is not necessarily a good indication of a vaccine efficacy, since the ability to block the infection of pathogens is related to the ability of the specific effector B and T cells in blocking the entry (by neutralizing antibody production) and/or controlling the spreading and replication of the pathogen (by killing the infected cells). A general activation of T cells does not provide information about the quantity and quality of the specific T cells. On the contrary it could create a very inflammatory environment that eventually may facilitate the pathogen entry (see HIV-1) or autoimmunity or other unwanted effects.
